# Influence of Using Food Delivery Applications on Adult Saudi Female Dietary Habits and Preferences during COVID-19 Lockdown Restrictions: Attitude Survey

**DOI:** 10.3390/ijerph191912770

**Published:** 2022-10-06

**Authors:** Reham M. Algheshairy, Raghad M. Alhomaid, Mona S. Almujaydil, Hend F. Alharbi, Woroud A. Alsanei

**Affiliations:** 1Department of Food Science and Human Nutrition, College of Agriculture and Veterinary Medicine, Qassim University, Buraydah 51452, Saudi Arabia; 2Department of Food and Nutrition, Faculty of Human Sciences and Design, King Abdulaziz University, Jeddah 21589, Saudi Arabia

**Keywords:** COVID-19 lockdown restrictions, food delivery applications, dietary habits, attitude, dietary behaviour

## Abstract

Food delivery applications (FDAs) shined during COVID-19 global lockdown restrictions. Consequently, lifestyle changes imposed a greater use of these applications over this period. These changes may strongly influence the nutritional health of individuals, particularly adult Saudi females. A cross-sectional study was performed to examine the influence of using FDAs during COVID-19 lockdown restrictions on attitude behaviours, including dietary habits and preferences among Saudi adult females. Participants voluntarily submitted their answers to a questionnaire administered via the Google Survey platform. Results illustrated that most Saudi female users of FDAs were aged between 18–24 years with 64.9%, 91.5% being single and 37% ordering food online within one to two days a month. There was a significant association between the influence of using FDAs during COVID-19 lockdown restrictions and age, education, and average days of ordering food online (*p* ˂ 0.05). Another important factor is that higher education was associated with more frequent use of the FDAs, there were direct relationships between education level and using FDAs, 58% of the participants were educated as undergraduate or postgraduate. Although lifestyle changes increased the use of FDAs during COVID-19 global lockdown restrictions, these changes may negatively affect individuals’ dietary habits and preferences, particularly adult Saudi females. These findings can aid in promoting healthy diet management globally and in Saudi Arabia unless the governments lead to significant beneficial changes toward improving food delivery applications.

## 1. Introduction

The coronavirus disease (COVID-19) pandemic began in Wuhan, China and spread rapidly worldwide in late December 2019 [1]. Since then, the discovered virus has caused severe medical complications and more than five million deaths worldwide [2]. Additionally, this pandemic has severely affected individual’s health and social status and disrupted the economy around the world. To minimise the epidemic’s risk, this has forced international governments to impose harsh lockdown rules on their citizens as a precautionary measure against spreading the infection [3,4]. Therefore, many food delivery apps (FDAs) in Saudi Arabia have become available 24 h a day, seven days a week, to allow customers to order from a wide array of foods that are extremely accessible [5]. In January 2019, the Saudi Food and Drug Administration issued a policy requiring all coffee shops and restaurants to include caloric information on their menus. However, this policy is not covered by all food delivery apps of these restaurants. Thus, acquiring accurate and adequate nutrition information in food delivery apps might raise consumer awareness and encourage them to consume fewer calories, which could help control Saudi Arabia’s obesity epidemic [6].

Another important aspect of the global lockdown is that all individuals’ daily interactions, including studying and working, were converted to digital services. This orientation is associated with reducing any threats to human health, decreasing the pressure on health systems, and preventing the collapse of the healthcare system [7,8,9,10]. It was strongly believed that many restaurants and cafes have contributed to the popularisation of FDAs as a trend to limit the spread of COVID-19, by minimising shared indoor space, maintaining physical distancing and adequate ventilation and taking caution when delivering [11]. FDAs allow users to browse a digital menu of restaurants and cafes via websites and smart devices [12]. However, these interactions with FDAs have negatively affected the population’s health and led to unhealthy behaviours, including sedentary behaviour and lack of physical activity [13,14]. In addition to the health status, FDAs have heavily affected individuals and communities’ shopping habits, food selection, nutritional content, and dietary intake [15]. For example, these applications can increase food consumption of high calories and low nutrient content of food, resulting in the risk of related diseases, including cardiovascular diseases, type 2 diabetes mellitus, and some cancers [16,17].

During the lockdown, HungerStation, Marsoul, and Jahez are the most popular online FDAs in the Arabian Gulf region, especially in Saudi Arabia. The success of these applications was obvious in delivering ready-to-eat meals to consumers and keeping food provider businesses viable [18]. Therefore, it has been hypothesised that increasing the use of FDAs during COVID-19 may exacerbate an individual’s dietary behaviour and preferences. No other data are currently available on the influence of using these applications on adult Saudi females’ attitudes, which affects the possible repercussions on their health and nutrition when purchasing food online. Therefore, this cross-sectional study examines the influence of using FDAs during COVID-19 between February and May 2021 on attitude, including dietary habits and preferences among adult Saudi females.

## 2. Materials and Methods

### 2.1. Study Subjects

A total of 3667 adult Saudi female volunteers took part in this study, aged 18 to 55 years old and a citizen living in different regions. The female subjects agreed to participate in the study by signing the consent form, and the Research administration at the Qassim university approved the study.

### 2.2. Study Design and Participants

A cross-sectional study was conducted in Saudi Arabia to examine the influence of using food delivery applications (FDAs) during COVID-19 global lockdown restrictions between February and May 2021. The inclusion criteria focused on female participants only. However, the exclusion criteria were under 18 and above 55 years old and missed any questions in the questionnaire. 

### 2.3. Questionnaire

Female participants voluntarily submitted their answers to a questionnaire administered via the Google Survey platform (Google LLC, 1600 Amphitheatre Parkway, Mountain View, CA 94043, USA). Specialists checked the questionnaire in related fields. External reviewers also provided feedback and recommendations for developing/improving the questionnaire. The experts were chosen and self-connected based on their expertise and publication of related works in nutrition. However, a number of adjustments were made to enhance the questionnaire’s reliability, validity, and scientific value of the data collected. So, to confirm the reliability and validity of the questionnaire, pilot research (*n* = 50 participants) was carried out. Cronbach’s alpha was obtained and found to be outstanding, reaching higher than 80% for measured values. 

The online questionnaire link was distributed via emails and familiar social media platforms such as WhatsApp, Twitter, Instagram, and Facebook. The survey included a cover letter in Arabic and comprised the following two sections: sociodemographic characteristics and health information. These sections consist of 25 closed-ended questions. The first part of the questionnaire was about demographics and health information, including age group; marital status (single, married, divorced, widowed); education level (high school completion or lower, diploma, bachelor’s degree, postgraduate degree); occupation (employed, unemployed, student); monthly income (≤1000SR (≤266 USD), >1000 to 5000 SR (>266 to 1330 USD), >5000 SR (>1330 USD), and region of residence (West, Middle, East, North, South). Health status questions include height and weight to determine the BMI level [19], suffering from any disease and obesity level.

The second part of the questionnaire referred to food delivery apps and included the following questions: On average, how much do you spend on food delivery apps per day? with the following responses (<50, 50−100, 100–200, >200 SR), Where do you often order your food? (House, Workplace, Other), How many times do you order food using food delivery apps? (Daily, 2–3 days a week, 3–4 days a month, 1–2 days a month), Have you ever used any of the following apps to order food? (Popular apps in Saudi Arabia). What are the payment methods you often use on delivery apps? -Responses were as follows: upon delivery, Mada, Apple pay, Visa. What is the main factor that determines your food choice using delivery apps? with the following responses (the restaurant, meal ingredients, meal picture, price, delivery fees, delivery time); what is the most important feature of food delivery apps? (delivery fees, delivery time, accept payment methods, user interface); does your mental state affect your food choice? (yes, no or sometimes); when do you prefer to order food? (morning, afternoon, evening). The respondents also answered questions about their behaviour towards meals, including knowledge of the ingredients; the number of calories in the meals or ordering high-calorie meals, as well as answering questions about their intention to minimise the use of food delivery apps in the future and the effects of their psychological state or the ability of advertisements for fast food to encourage them to use food delivery apps. The last question was about the impact of the corona pandemic on the use of food delivery apps? With the following responses (use increased, use decreased, no change).

### 2.4. Sample-Size Calculation

This study calculated the sample size using the Raosoft online calculator (Raosoft Inc., Seattle, WA, USA, http://www.raosoft.com/samplesize.html, accessed on 3 January 2021). The minimum number of participants to be included in this study is 385 after assuming the margin of error and confidence level to be 5% and 95%, respectively, with a reported female population of approximately 14.7 million (SA Statistics Bureau, 2021, https://www.stats.gov.sa/en, accessed on 2 January 2021). However, to reduce errors and obtain more accurate results, the sample size for the current study was 3667 participants.

### 2.5. Statistical Analysis

Statistical analysis of the results was carried out using the Statistical Software for the Social Science version 25.0 (SPSS Inc. Chicago, IL, USA), at the significant level of *p*-value < 0.05. Data are presented as frequency and percentage for categorical variables and mean with standard deviation for numerical variables. The chi-square statistical test is used to determine whether there is a relationship between demographic characteristics and the food delivery application user’s behavioural pattern.

## 3. Results

### 3.1. Sociodemographic Characteristics

In the four months of data accumulation, a total of 3667 responses were received from adult Saudi female participants who completed the survey. Table 1 summarises the overall sociodemographic characteristics of respondents. After analysing the data, most participants were aged between 18 and 24 (64.9%). Most participants were single (91.5%), and only 8.4% were married. From an educational point of view, 58.3% of participants were undergraduate and postgraduate degrees; while most participants were students (82.2%); the monthly income ranged from less than or equal to 1000 SR/month (80.5%) to more than 5000 SR (4.9%). The participants in this survey were from different regions of Saudi Arabia (Middle, West, East, North and South) 50.8%, 21.6%, 11,9%, 8.1% and 7.5%, respectively. Regarding general health status, most participants did not suffer from diseases (93.3%) such as diabetes, anemia, asthma, blood pressure, and heart disease. Eight participants answered not obese when asked in this survey, do you consider yourself obese? However, heights and weights were obtained from 3307 participants, where a mean ± standard deviation was reported as 158.2 ± 5.9 cm for height, 55.9 ± 12.7 kg for weight, and then BMI was calculated as 22.3 ± 4.8 kg/m^2^, as weight/height^2^. According to the BMI measurement, more than half of the participants were classified as normal weight by BMI 18.5–24.9, and the rest were classified as underweight, overweight, and obese 19.7%, 15% and 7.6%, respectively.

### 3.2. Prevalence and Frequency of Using Food Delivery Applications and the Sociodemographic Characteristics of Adult Saudi Female User’s Patterns during COVID-19 Lockdown Period

Summarised responses to using food delivery applications (FDAs) on the prevalence and dietary patterns among adult Saudi females have presented in Table 2. Generally, the three most popular FDAs were HungerStation, Marsoul and Jahez in Saudi Arabia. While most apps were used (%) during COVID-19 lockdown restrictions as follows: Hunger-Station and Marsoul (30%), followed by HungerStation, Marsoul and Jahez (14.6%), HungerStation only (14.2%), Marsoul only (7.6%) and finally by Hunger-Station, Marsoul, Jahez and To you (6.3%). A third of participants (38%) reported using debit cards (Mada, Visa, Mastercard) with Apple Pay on delivery. In comparison, less than one-quarter of participants used cash on delivery (21.7%), and the rest used debit cards on delivery (Mada, Visa, Mastercard) (18.4%). The average user’s spending per online order from FDAs is 45.1% and 44.3% for at least 50 SR and between 50–100 SR, respectively. From a monthly income perspective, it is imperative to point out how many days a participant order food through apps. The results showed that participants were classified based on their food online orders within one to two days a month, a few days a month, a few days a week and daily with 37.6%, 36.3, 24.3% and 1.7%, respectively. Most food orders (91.2%) were placed from home, while only 2% and 6.7% were sent from the workplace and other places. However, participants were asked whether to get their order as a takeaway delivery or go to restaurants and cafes. More than half of the participants (57.4%) used both ways of receiving their food orders. The most preferred time to order food through apps was dinner time (49.1%) and midnight (30.2%). The top four factors affecting the participants’ meal choice through FDAs were the restaurant with the highest score 39%, followed by delivery fees with 18.3%, the meal ingredients with 17.8%, and the meal price with 17.6%. 

### 3.3. The Influence of Caloric Information Posting in Food Delivery Applications on Participants’ Decision to Select Their Meals 

Two-thirds of the participants (60%) reported that they did not pay attention to the calories of the whole meal written on the menu, and 26.7% of the participants responded sometimes. Only 13.1% paid attention to calories on the menu. When ordering high-calorie meals, approximately half of the participants (53.6%) reported not paying attention to calorie information for ordering high-calorie meals from FDAs during the COVID-19 lockdown period. The rest of the participants responded with sometimes for paying attention to high-calorie meals (36.3%), while 10.1% of participants used this information to decide what to order. Participants were divided about their concerns about the meal ingredients regarding the number of calories that would improve their decision to select lower-calorie meals. Roughly two-thirds of participants (60.1%) reported that meal ingredient posting would not influence their purchase. In contrast, 20.5% of participants reported that sometimes they read the description of the meal ingredients to select the lower caloric food choices, and approximately 18.1% of participants reported that they paid more attention to meal ingredients posted on the FDA menu. 

### 3.4. The Influence of Fast-Food Advertising in Social Media Platforms on Using Food Delivery Applications and Changing Participants’ Eating Patterns 

At the point of fast-food online purchase, the results reveal to what extent fast-food advertising on social media platforms targets the emotional state of consumers who order and consume these meals from food delivery applications. More than half of the participants (58.8%) reported that they were greatly influenced by fast-food advertising on social media platforms and motivated to order high-caloric meals using FDAs compared with those who were not affected. Moreover, almost a third of the participants (31.8%) changed their eating patterns after using FDAs. Nearly half of the participants (49%) selected the delivery charge of FDAs compared to delivery speed (27.3%). Around 42.3% of participants reported that their psychological state affected the type of meal ordered online. More than half of the participants (55.4%) intend to reduce the use of FDAs in the future.

### 3.5. Association between Sociodemographic Characteristics and Changing Participants’ Eating Patterns When Using Food Delivery Applications 

A cross-sectional analysis of the association between sociodemographic characteristics (age group, BMI, education level and income) and changing participants’ eating patterns when using food delivery applications (FDAs) during the COVID-19 lockdown period was carried out. A significant association was found between the participant age group, BMI, education level, income, and some questions presented in Table 3. These questions include the main factor determining your meal choice (type of restaurant, meal ingredients, meal picture, meal price and so on), do you avoid ordering high-calorie meals?, has your eating pattern changed after using FDAs?, and the most important feature to use food delivery applications. However, no meaningful effect on observed associations was found between each age group, BMI, education level, income, and the question about fast-food advertising on social media platforms.

### 3.6. Influence of COVID-19 Pandemic on Using Food Delivery Applications

Figure 1 illustrates the general usage of food delivery applications (FDAs) among adult Saudi females during the COVID-19 pandemic. Almost half of the participants increased their FDAs’ usage by 46.2%, followed by 16.2% who decreased their usage, but 37.6% did not change their usage behaviour during the pandemic. A significant association has been found between the influence of using FDAs during the COVID-19 pandemic and age, education, perceived obesity status, average days of ordering food, and avoidance of high-calorie food, as shown in Table 3.

## 4. Discussion

The use of Food Delivery Applications (FDAs) expanded greatly during COVID-19 global lockdown restrictions. Consequently, this pandemic has changed the attitudes of individuals towards the greater use of these applications. This paper, therefore, aimed to examine the influence of using FDAs on attitude, including dietary habits and preferences among adult Saudi females. This cross-sectional study was conducted on 3667 adult Saudi females during COVID-19 lockdown restrictions and found that the majority (91.2%) usually ordered food from FDAs while staying home. One may believe that this period can alter their dietary habits when forcing most individuals to remain at home for a prolonged period, with unlimited ease of access to palatable food. The study findings show that this belief is consistent with previous studies on ease-of-access, ease-of-use and speed-of-delivery of food delivery applications as important factors that impact prevalence, attitude and dietary behaviour during the pandemic [21]. 

### 4.1. Prevalence and Frequency of Using Food Delivery Applications and the Sociodemographic Patterns among Adult Saudi Female Users during COVID-19 Lockdown Period 

To date, the prevalence and frequency of using food delivery applications and the sociodemographic patterns among adult Saudi female users during the COVID-19 lockdown period have not been investigated. Therefore, current understanding of the links between the prevalence and frequency of using FDAs among those users and their sociodemographic characteristics (age, education level, marital status, occupation, perceived obese status and income) and their habit patterns (i.e., do you use FDAs, how often FDAs are used, payment method used, spending rate, place of order, ways of order receiving, time of day to order, factors influencing an ordered meal from FDAs) will establish a baseline for comparison during and after the pandemic. Among the study participants, 44.6% assert a continued usage of FDAs in the future. Similarly, Stephens et al. [12] confirmed that food delivery services had been preferred, especially during dinner. Adopting and maintaining healthy lifestyle habits when using FDAs is recommended as a fundamental health principle. Thus, analysing their dietary habits, preferences and frequency using FDAs (72.7%) can express the future predominance of health risk behaviours such as high body mass index [22]. These health risks may become more prevalent and could persist after the pandemic among those students (82.2%) receiving monthly financial rewards (80.5%), aged between 18–24 years (64.9%), with either high school and lower qualifications (41.7%), or undergraduate and postgraduate degrees (58.3%), specificity for those who were not self-perceived obese (80%). For this reason, such a study allows future interventions to be targeted toward frequent users to serve as an indicator of potential public health issues and the need for further research on improving the content and services of FDAs.

### 4.2. Marital Status

Most participants were single (91.5%), and only 8.4% were married. Most participants who were unmarried were more influenced to use FDAs and desired to increase their caloric intake than their counterparts. One possible perception could be marital status. The single women found it easy to use FDAs. Their responsibility toward meal planning and home-meal preparation appeared to be poor compared to married ones, who most likely reflect more interdependent self-orientation with their families for meal planning and preparation [23]. Another study referred to the decrease in the online food industry during the COVID-19 pandemic as anxiety about food hygiene from outside [5]. 

Single women find it easy to use apps, while married women are mostly committed to preparing meals for their families. For this reason, the percentage of obese single participants is higher than the married participants. The total percentage of adult Saudi female participants who were obese (n= 733, 20%), single subjects, and married were 17.2% and 2.8%, respectively. 

One more finding of this study is that unhealthy eating patterns are prevalent among adult Saudi females by ordering from food delivery applications. Prevalence of such behavioural traits significantly counts toward overweight and obesity, provoking the need for improvements in FDAs and obese management strategies by partners, healthcare providers and stakeholders [24]. 

### 4.3. Time of Day to Order Using FDAs

The results in the present study express the vital role of FDAs during COVID-19 lockdown restrictions. 72.7% of the participants preferred to order meals using FDAs. Generally, the conventional time for dinner in Saudi Arabia is between 8:00–9:30 pm [25]. However, findings showed that the preferred time to order is at dinner and midnight, with 49.1% and 30.2%, respectively. The negative changes in the majority of late-night-dinner eating behaviour could attribute to the COVID-19 pandemic, including prolonged home confinement, social distancing, isolation, being up late at night, psychological status, feeling a sense of boredom and stress [15,26,27,28,29,30], supported by the increased availability of food and meal options and differential pricing offered by FDAs [31]. It is recommended to adhere to regular meal schedules to maintain a balanced diet and create a more stable energy source and continuous daily physical activity. 

### 4.4. High-Calorie Food Choices Using FDAs

FDAs pose an inevitable challenge to the public health system by promoting unhealthy eating habits. It does this by enlarging the range of high-calorie food choices. Findings found roughly 32% of the respondents admitted that FDAs have greatly affected their eating habits to COVID-19 compared to those who did not change their eating patterns by ordering wisely and eating healthy meals when using FDAs (68.2%). During the confinement, the majority of participants were single (91.5%), students of distance learning from home (82.2%), receiving monthly financial rewards from their institutions (80.5%) and using FDAs (72.7%), which made it difficult to cook and consume healthy homemade meals. Besides, these factors may affect 53.6% of participants to be highly motivated toward consuming high-calorie fast foods. Other studies showed similar findings about the increased intake of carbohydrates, fat and fried food consumption [5,32]. Therefore, unhealthy food consumption paired with low physical activity, especially for students with time constraints or busy schedules, may yield a positive energy balance, increased body weight and related chronic diseases [22,33,34,35,36,37,38,39,40,41,42]. Similar findings concerning the increased consumption of high calories due to COVID-19 confinement were also observed [24,43,44,45]. In 2004, a brief review was published on the prevalence of physical inactivity in Saudi Arabia, indicating that most Saudi females are not physically active at a sufficient level (intensity, duration, frequency) [36]. In order to counteract poor dietary behaviours, meal planning and controlling food composition and meal caloric content, adult females should be targeted for interventions aimed at maintaining and improving physical activity and dietary practices during this pandemic and beyond. 

### 4.5. Fast-Food Advertising on Social Media Platforms and FDAs

During the lockdown restrictions, fast food advertising on social media platforms targeting all age groups, especially adults, has grown exponentially [46]. In this study, the descriptive analysis results highlighted that almost half of the participants (58.8%) agreed that social media platforms encouraged fast-food consumption through the use of FDAs. Not only that but also 42.3% of participants agreed that their psychological state affected their meal type when using FDAs. While a total of 57.7% of the participants responded (28.9% indicated not affect and 28.8% indicated sometimes affect) about the impact of their psychological state on the type of meal chosen. This pattern has a close similarity to that found by [47], which is that social media platforms have a great effect on the mental (psychological) state of a customer when choosing food [47]. Presumably, the reason for this could be unhealthy food marketing through the leading social media platforms that have expanded to incorporate live streaming and augmented reality. In contrast, 41.2% of the participants were not influenced by social media. Affecting their usage can be related to their awareness of healthy and unhealthy food advertising and the reliability issues on social media [48,49]. This effect is an important indicator for both groups that cannot be ignored because this period would significantly impact an individual’s nutritional status. However, there is no implemented ban, prohibition, or restriction on high-caloric food advertising on social media platforms or the FDAs globally and in Saudi Arabia. Therefore, globally fast-food services do not respond effectively to their health ministries’ calling for reducing high-energy foods, especially in their digital menus [50,51,52]. Unless the government-led changing approach effectively regulates social media advertising to protect public health. This contribution can be achieved by encouraging healthy food providers to participate in FDAs for a small fee, providing attractive pictures of healthy dishes and reviewing their prices because they are often expensive compared to unhealthy, especially high-calorie meals. The governmental supervisory approach, as a third-party e-commerce platform, should mitigate the negative and promote the positive impacts of online food delivery on users’ health, thus ensuring its sustainability.

### 4.6. Influence of Food Calorie Information on Using FDAs

As a part of Saudi Arabia’s vision of 2030 to promote a healthier lifestyle, the Food and Drug Authority in Saudi Arabia obligated all food establishments to apply food calorie information on digital and non-digital menus by the end of 2018. The Ministry of Municipal Rural Affairs and Housing will be responsible for monitoring, and the Food and Drug Authority will be responsible for ensuring the correctness of calculating calories. A violation ticket for non-compliance will be issued to food businesses. Despite Saudi Arabia’s keenness to offer health protection and care to citizens, 60% of participants in this study neglected the importance of calories displayed on the FDA’s menu. In contrast, 26.7% of participants reported sometimes being influenced by the number of calories when ordering meals, and 13.1% of participants paid attention to such information. According to multiple previous studies, opposite opinions were observed that females were affected mainly by calorie labelling provided by restaurants than their counterparts [53,54,55]. Additionally, the findings of this study suggest that calorie information did not change the participant’s habits even when the meal ingredients were displayed on menus (61.5%). Other reviews emphasised a similar opinion that the effectiveness of calorie labelling on the menu has little impact on purchasing fast-food meals [56,57,58,59]. However, there is a lack of information on the influence of food calorie labelling on the menu of FDAs. Whereas Girz et al. [60] found it is not promising that providing calorie labelling on the menu can prevent obesity. This emerging evidence shows that calorie information and meal ingredients (content) on the restaurant’s digital and non-digital menus did not similarly influence food ordering decisions of all population groups. Thus, it may have unintended consequences for individuals who struggle with disordered eating or other weight-related concerns. 

### 4.7. Strength and Limitations

The study’s results supported behavioural changes regarding using FDAs during the lockdown restriction, which might be used to create health and dietary initiatives that support healthy habits. However, the online-based survey may limit people participating as they have limited or no access to the internet leading to the possibility of sample bias.

## 5. Conclusions

Although lifestyle changes imposed a greater use of food delivery applications (FDAs) during COVID-19 global lockdown restrictions, these changes may reflect negatively associated with individuals’ dietary habits and preferences, particularly adult Saudi females. These findings can aid in promoting healthy diet management globally and in Saudi Arabia unless the governments lead to significant beneficial changes toward improving food delivery applications. Additionally, future research is recommended to maximise the positive and reduce the adverse effects of using FDAs by accessing healthy foods for all populations during and post-pandemics. Thus, to succeed, all partners, healthcare providers and stakeholders, including online food delivery app creators, providers, policy-makers, users, and academic research and development (R&D), should work together to increase the attention and awareness of healthy eating.

## Figures and Tables

**Figure 1 ijerph-19-12770-f001:**
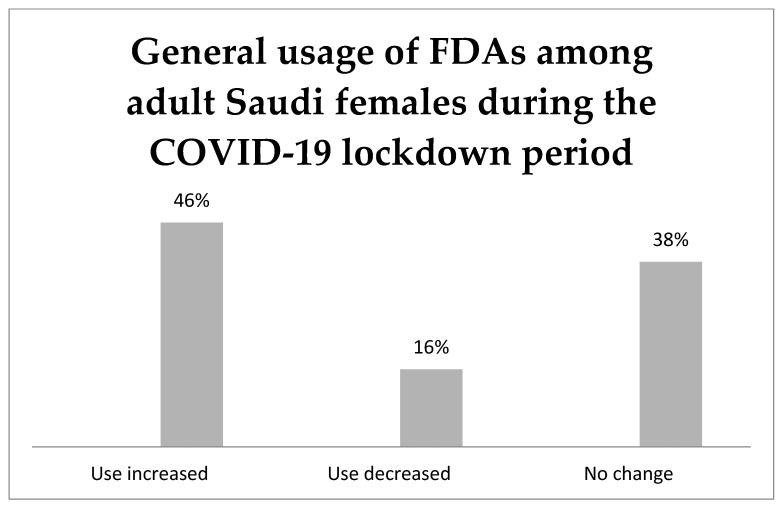
General usage of food delivery applications (FDAs) among adult Saudi females during COVID-19 lockdown period represented 46, 38 and 16% of increased, no change, and decreased their FDAs’ usage, respectively.

**Table 1 ijerph-19-12770-t001:** Sociodemographic and personal characteristics of the participants (n = 3667).

Statement/Question	Variable(s)	Number/Mean ± SD	Percent (%)
Age group (years)	18 years	838	22.9
	>18–24	2380	64.9
	25–34	393	10.7
	35–44	50	1.4
	45 to 55	6	0.2
Nationality	Saudi	3667	100
Education level	High school and lower qualification	1528	41.7
	Undergraduate and postgraduate degrees	2139	58.3
Marital status	Single	3358	91.5
	Married	309	8.4
	Divorced/widowed	0	0
Occupation	Employed	138	3.8
	Unemployed	516	14.1
	Student	3013	82.2
Monthly income Saudi Riyals (SR)	Less than or equal to 1000	2953	80.5
	More than 1000 to 5000	536	14.6
	More than 5000	178	4.9
Region of residence	Middle	1862	50.8
	West	793	21.6
	East	438	11.9
	North	297	8.1
	South	276	7.5
Do you suffer from any disease?	No	3421	93.3
	Diabetes	48	1.3
	Anaemia	66	1.8
	Asthma	71	2
	Blood pressure	35	1
	Heart disease	22	0.6
Do you consider yourself obese?	Yes	733	20
	No	2934	80
BMI group N= 3507	<18.5	692	19.7
	18.5–24.9	2024	57.7
	25.0–29.9	526	15
	>30	265	7.6
Body Measurements	Weight (Kg)	55.9 ± 12.7	
	Height (cm)	158.2 ± 5.9	
	BMI	22.3 ± 4.8	
Total answer		3507	95.6
Total no answer		160	4.36

**Table 2 ijerph-19-12770-t002:** Assessment of responses to the questionnaire questions among the participants (n = 3667) during COVID-19 lockdown restrictions.

Questions	Variable(s)	Number of Responses	Percent (%)
Do you ever use any of the food ordering apps during COVID-19 lockdown?	Yes	2666	72.7
	No	1001	27.3
Have you ever used any of the food ordering apps during COVID-19 lockdown? (mostly)	via HungerStation and Marsoul	1099	30
	via HungerStation, Marsoul and Jahez	537	14.6
	via HungerStation only	522	14.2
	via Marsoul only	277	7.6
	via HungerStation, Marsoul, Jahez and To you	231	6.3
What payment methods do you use when ordering food through apps? (mostly)	Cash on delivery	795	21.7
	Debit cards (Mada, Visa, Mastercard) on delivery	677	18.4
	Debit cards (Mada, Visa, Mastercard) with Apple Pay on delivery	1393	38
	Cash on delivery	795	21.7
What is your average spending per order using food ordering apps?	<50 SR	1625	45.1
	50–100 SR	1652	44.3
	>100 SR	344	9.4
	>200 SR	46	1.3
On average, how many days do you order food through apps?	Daily	64	1.7
	A few days a week	892	24.3
	7 to 10 days a month	1331	36.3
	1–2 days a month	1380	37.6
Where do you usually order from food ordering apps?	House	3345	91.2
	Workplace	75	2
	Other	246	6.7
Do you get your order as a takeaway delivery or go to restaurants and cafes?	Ordering food delivery (takeaway)	995	27.1
	Go to restaurants and cafes (collecting myself)	566	15.4
	Both	2106	57.4
What is your preferred time to order food?	Morning	170	4.6
	Afternoon	159	4.3
	Evening	430	11.8
	Dinner time	1802	49.1
	Midnight	1106	30.2
What is the main factor determining your meal choice through food ordering applications?	Type of restaurant	1431	39
	Meal ingredients	654	17.8
	Meal picture	175	4.8
	Meal price	644	17.6
	Delivery fees	672	18.3
	Delivery time	91	2.5
When choosing a meal, do you pay attention to the number of calories written on the restaurant menu?	Yes	482	13.1
	No	2205	60.1
	Sometimes	980	26.7
Do you avoid ordering high-calorie meals?	Yes	370	10.1
	No	1967	53.6
	Sometimes	1330	36.3
When you choose a meal, do you read the description of the meal ingredients regarding the number of calories?	Yes	662	18.1
	No	2254	61.5
	Sometimes	751	20.5
Is fast-food advertising in social media platforms encouraging you to order from food delivery applications?	Yes	2157	58.8
	No	1510	41.2
Has your eating pattern changed after using food delivery applications?	Yes	1167	31.8
	No	2500	68.2
What is the most important feature of using food delivery applications?	Application service quality	545	14.9
	Delivery charge	1796	49
	Delivery speed	1000	27.3
	Ease of payment	326	8.9
Does your psychological state affect the type of your meal?	Yes	1552	42.3
	No	1058	28.9
	Sometimes	1057	28.8
Do you intend to reduce your use of food delivery applications in the future?	Yes	2033	55.4
	No	1634	44.6

Mada: the second e-payment system in Saudi Arabia, operates through a network of global technical payment systems [20].

**Table 3 ijerph-19-12770-t003:** Exploring the relationship between sociodemographic characteristics (age, education level, income and BMI group) with selected questions on using food delivery applications (FDAs) and participant’s dietary behaviour and preferences using Chi-Square test’s *p*-value.

Questions	Age	Education Level	Income	BMI Group
	Chi-Square Test *p* Value			
Do you consider yourself obese?	0.000	0.003	0.761	0.000
What is the main factor determining your meal choice through food ordering applications?	0.002	0.000	0.000	0.071
Do you avoid ordering high-calorie meals?	0.000	0.000	0.000	0.000
When choosing a meal, do you pay attention to the number of calories written on the restaurant menu?	0.000	0.000	0.006	0.000
Is fast-food advertising in social media platforms encouraging you to order from food delivery applications?	0.247	0.359	0.839	0.364
Has your eating pattern changed after using food delivery applications?	0.000	0.000	0.001	0.000
What is the most important feature of using food delivery applications?	0.000	0.001	0.000	0.001
Does your psychological state affect the type of your meal?	0.001	0.000	0.300	0.000

*p*-value < 0.05 indicated a significant association of the variables with the responses provided by participants to the questions.

## Data Availability

The corresponding author can provide all the data used in the present study upon reasonable request.

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
