# Peer review of "Influence of Using Food Delivery Applications on Adult Saudi Female Dietary Habits and Preferences during COVID-19 Lockdown Restrictions: Attitude Survey"

_ijerph, 2022, doi:10.3390/ijerph191912770_

Round 1
Reviewer 1 Report
Dear Authors,
This is a very interesting and timely paper.
The following revisions are recommended for improvement before publication:
Abstract:
lines 15-16 and general: Authors wrote: „This cross-sectional study aims to examine the influence of using FDAs during COVID-19 lockdown restrictions on knowledge and attitude lifestyle behaviours,…” but I wonder if rather the knowledge can influence the use of FDAS? Could Authors comment on this?
The authors write about the relationship between certain discriminants and the FDAs. At this point, however, it is worth mentioning the direction of relationships, e.g. was higher education associated with more frequent use of the FDAs?
Introduction:
In this section, the authors should better describe the FDAs available and popular in the country. Is it a kind of diet catering or rather applications that facilitate the purchase of the menu of popular restaurants? Do these apps also provide nutritional information? In this case, their use may affect nutritional knowledge. Otherwise, they may rather affect eating behavior / preferences but not knowledge. In this section, the authors should also explain why only women were included in the study.
Materials and methods:
line 88: For the convenience of the reader, authors should rather operate in relation to income to the national average or minimum wage. Then the foreign reader can better understand the context.
line 90: BMI level is to define to body weight status, including obesity levels.
Did the authors validate the questionnaire? Did the authors introduce criteria for including and excluding people from the study?
Have all completed questionnaires been included in the analysis? If not, what was the rejection rate? What BMI criteria were adopted to assess the correctness of the body weight? This information is missing.
Sample-size calculations: Despite the larger study group, it is not representative due to the method of acquiring respondents.
Results:
Table 1. It is not necessary to show the gender of the respondents in the table if only women were included in the study.
Criteria for education level are unclear in the table 1. E.g. what Authors ment by „student” category?
The authors did not precisely define the age category: e.g. a category> 18 means that the questionnaire could be completed by children. In this case, usually the consent of the guardian is needed. Rather, minors should be excluded from the study. Also, the category 45-55, due to the very small number of respondents (6 people), cannot be included in the statistical analysis.
Table 2: Do the authors analyze the impact of the type of payment method? This is probably not related to the topic and does not provide relevant information. Likewise, the name of the application has no meaning to the reader outside the country, because the reader is unfamiliar with these applications and does not know what they are different (if they are different). This information is only relevant for readers from the country where the survey was conducted.
Discussion:
I like breaking the discussion into sections, but the discussion itself is more of a discussion of the results. It is worth referring to other works more.
General: Authors refer their results to adults (e.g. line 387), but there are respondents age below 18? What is the age category for adults in Saudi Arabia?
Authors should describe the limitation of their study!
Author Response
Dear reviewer(s),
We appreciate the time devoted and the proposals you have made. The revised version of the manuscript has been modified according to your comments. Your detailed argumentation of each of the points covered in your review has allowed us to realize some deficiencies when transmitting the work done and improve it for the reader. We will be at your disposal to make any change that you deem necessary, resolve any questions or proceed with new revisions.
In the table below, all comments are answered. The lines in the “reviewer’s comment” and “authors’ reply” have been written with the “track changes mode in MS Word” activated.
Responses to Reviewer 1
Reviewer’s comment 1 on Abstract |
Original version line(15-16): |
Authors wrote: „This cross-sectional study aims to examine the influence of using FDAs during COVID-19 lockdown restrictions on knowledge and attitude lifestyle behaviours,…” but I wonder if rather the knowledge can influence the use of FDAS? Could Authors comment on this? |
|
Author’s reply |
Revised version line(s): 17-18 |
During Lockdown restrictions, many people tend to use FDAs as a simple way to get food rather than their knowledge of products or their attitude or lifestyle behavior. So, it has been aimed that the influence of using FDAs during lockdown restriction on knowledge and attitude lifestyle behavior |
|
Reviewer’s comment 2 on Abstract |
Original version line(s): |
The authors write about the relationship between certain discriminants and the FDAs. At this point, however, it is worth mentioning the direction of relationships, e.g. was higher education associated with more frequent use of the FDAs? |
|
Author’s reply |
Revised version line(s): 25-27 |
Another important factor that higher education was associated with more frequent use of the FDA's, there were direct relationships between education level and using FDAs, 58% of the participants were educated as an undergraduate or postgraduate |
|
Reviewer’s comment 3 |
Original version line(s): |
Introduction:In this section, the authors should better describe the FDAs available and popular in the country. Is it a kind of diet catering or rather applications that facilitate the purchase of the menu of popular restaurants? Do these apps also provide nutritional information? In this case, their use may affect nutritional knowledge. Otherwise, they may rather affect eating behavior / preferences but not knowledge. In this section, the authors should also explain why only women were included in the study. |
|
Author’s reply |
Revised version line(s): 46-59 |
It has been added to the introduction
|
|
Reviewer’s comment 4 |
Original version line(s): 88 |
Materials and methods: line 88: For the convenience of the reader, authors should rather operate in relation to income to the national average or minimum wage. Then the foreign reader can better understand the context. |
|
Author’s reply |
Revised version line(s): 123 |
monthly income (≤1000SR (≤266USD), >1000 to 5000SR (>266 to 1330USD), >5000SR (>1330USD),……. |
|
Reviewer’s comment 5 |
Original version line(s): 90 |
Materials and methods: line 90: BMI level is to define to body weight status, including obesity levels. |
|
Author’s reply |
Revised version line(s): 126 |
Yes, it is. According to (Kruizenga, Hofsteenge et al. 2016) |
|
Reviewer’s comment 6 |
Original version line(s): 80 |
Materials and methods: Did the authors validate the questionnaire? Did the authors introduce criteria for including and excluding people from the study? |
|
Author’s reply |
Revised version line(s): 107-114 |
More details have been given in that section.
|
|
Reviewer’s comment 7 |
Original version line(s): |
Materials and methods: Have all completed questionnaires been included in the analysis? If not, what was the rejection rate? What BMI criteria were adopted to assess the correctness of the body weight? This information is missing. |
|
Author’s reply |
Revised version line(s): |
For the BMI of respondents, we calculated it using the given height and weight using the common equation of BMI, then the nutritional status was classified using the known scale (Kruizenga, Hofsteenge et al. 2016) |
|
Reviewer’s comment 8 |
Original version line(s): |
Materials and methods: Sample-size calculations: Despite the larger study group, it is not representative due to the method of acquiring respondents. |
|
Author’s reply |
Revised version line(s): 459 |
This has been involved in limitation of the study. |
|
Reviewer’s comment 9 |
Original version line(s): |
Results:Table 1. It is not necessary to show the gender of the respondents in the table if only women were included in the study. |
|
Author’s reply |
Revised version line(s): |
Gender has been deleted from table 1.
|
|
Reviewer’s comment 10 |
Original version line(s): |
Results:Criteria for education level are unclear in the table 1. E.g. what Authors ment by „student” category? |
|
Author’s reply |
Revised version line(s): |
That means they are still student either undergrad or postgrad.
|
|
Reviewer’s comment 11 |
Original version line(s): |
Results: The authors did not precisely define the age category: e.g. a category> 18 means that the questionnaire could be completed by children. In this case, usually the consent of the guardian is needed. Rather, minors should be excluded from the study. |
|
Author’s reply |
Revised version line(s): |
The questionnaire it was for female age18 years and over, it was stated in study subject "A total of 3667 adult Saudi female volunteers was enrolled in this study with ages from 18 to 55 years old" we might include clear state in methods as " The inclusion criteria were focusing on female participants only. However, the excluding criteria were under 18 and above 55 years old as well as miss responded any question in the questionnaire. "
|
|
Reviewer’s comment 12 |
Original version line(s): |
Results: Also, the category 45-55, due to the very small number of respondents (6 people), cannot be included in the statistical analysis. |
|
Author’s reply |
Revised version line(s): |
It has been reanalysis the statistical analysis again without this category 45-55 age in table 3 |
|
Reviewer’s comment 13 |
Original version line(s): |
13 a . Table 2: Do the authors analyze the impact of the type of payment method? This is probably not related to the topic and does not provide relevant information. 13 b . Likewise, the name of the application has no meaning to the reader outside the country, because the reader is unfamiliar with these applications and does not know what they are different (if they are different). This information is only relevant for readers from the country where the survey was conducted. |
|
Author’s reply |
Revised version line(s): |
13a . Yes, this is not related to the topic as we present mostly in table 2
13b. We have been included in this study to be specific and accurate information as some of the has some features like (price, quick delivery) |
|
Reviewer’s comment 14 |
Original version line(s): |
Discussion: I like breaking the discussion into sections, but the discussion itself is more of a discussion of the results. It is worth referring to other works more. |
|
Author’s reply |
Revised version line(s): |
It has been done |
|
Reviewer’s comment 15 |
Original version line(s): |
General: Authors refer their results to adults (e.g. line 387), but there are respondents age below 18? What is the age category for adults in Saudi Arabia? |
|
Author’s reply |
Revised version line(s): |
It has been modified the age in the manuscript |
|
Reviewer’s comment 16 |
Original version line(s): |
Authors should describe the limitation of their study! |
|
Author’s reply |
Revised version line(s): 459 |
The limitation has been added. |

Reviewer 2 Report
Language and technical care:
The manuscript requires some major attention in terms of overall language and technical aspects, with a few examples highlighted below:
- Line 36 – remove the word ‘the’;
- Line 38 – replace the words ‘have been approved’ with ‘become a norm’;
- Line 39 – replace ‘bases have’ with interactions;
- Line 40 & 41 – these sentences need rephrasing;
- Line 40 – consider changing ‘to save the’ to ‘ensure possible safety from’;
- Line 47 – replace ‘rules’ with ‘interactions with FDAs’;
- Line 48 – check for consistency in the use of ‘behaviours’ and ‘behaviors’;
- Line 52 – add the words ‘of food’ after nutrient content;
- Line 55 – begin the sentence with ‘During lockdown, HungerStation…;
- Line 57 & 58 – this sentence needs total rephrasing;
- Line 70 – replace ‘was enrolled’ with ‘took part’;
- Line 71 – insert the word ‘and’ between ‘old’ and ‘a citizen’;
- Line 81 – rephrase – ‘links that were distributed via emails and familiar…’;
- Line 82 – remove ‘design was’;
- Line 84 – end sentence after ‘questions’, start new sentence with ‘The first part…’;
- Line 90 – ‘height’ not ‘high’;
- Line 90 – ‘determine’ not ‘define’;
- Line 99 – on page 5 reference is made to the payment facility ‘Mada’, here it is referred to as ‘Made’;
- Line 101 – Meal ingredients, Meal’s picture – check for consistency in the use of caps and punctuation marks;
- Line 113 – beginning bracket that should attach to the words ‘hard break’;
- Line 127 – add the word ‘behavioural patterns’ to read ‘user’s behavioural patterns’;
- Line 135 – replace ‘were’ with ‘had’;
- Line 136 – the percentage of the ‘single’ criteria of sample is repeated in one sentence;
- Line 141 – remove the word ‘were’;
- Line 152 to 157, as well as the variables in Table 2 on page 6 – the authors should take note to inform the reader of how HungerStation – Marsoul relates to HungerStation – Marsoul – Jahez, versus ‘HungerStation Marsoul’, particularly in Table 2, where it feels as if there is duplication – if the name of these APPS are indeed preceded by HungerStation in each example, then the authors should indicate it as such;
- Line 161 – ‘using’ to replace ‘used’;
- Line 166 – what is the purpose of the word ‘often’?
- Line 201 – what does ‘interesting feature’ mean?
- Page 6 – Table 2 – in the second, third and fourth column of the question ‘On average, how many days do you order food through apps?’ there is an open line – remove;
- Page 7 – Table 2 – what does the word ‘interesting’ imply in the most ‘interesting feature to use food delivery apps?
- Line 240 – begin sentence with ‘The use of Food Delivery Applications (FDAs) expanded greatly during the COVID-19…’;
- Line 279 – 283 – sentence too long;
- Line 286 – Rather ‘Single women find it easy’;
- Line 289 – full-stop should be replaced by comma;
- Line 291 – rephrase ‘are well widespread’ is not correct;
- Line 345 – replace ‘did not’ with ‘were not’;
The manuscript is very well referenced using relevant, up-to-date references, however the reviewer is uncertain if the in-text manner of referring to authors (replacing it by a bracketed number as in line 378) is correct.
Literature Review:
This reviewer felt that better explication of what knowledge is, what food knowledge is, what dietary knowledge is, and particularly what attitudes are, should be provided. The authors make reference to a concept referred to as ‘attitude lifestyle behaviours’. This reviewer could find no such concept. This is concerning, as ‘attitudes’ are a very specific concept with very specific dimensions and indicators to measure.
In fact, when using Google Scholar, the reviewer found this exact paper available as follows: https://assets.researchsquare.com/files/rs-1467008/v1/2fb4dbb1-833c-4b38-9250-9b9177f8d0a4.pdf?c=1656577778
Methodology and materials:
The reviewer believes that the methodology is straightforward and explained well, and that good and acceptable research procedures have been followed.
Results and Discussion:
The reviewer believes that the results and discussion of the results are all scientifically sound, but that the presentation of the results, specifically the sentence construction when the results are discussed, makes for very difficult reading.
In line 19, the authors indicate that the sample was between 18 – 24 years, however in table 1, a large proportion of the sample (23%) were UNDER 18 years old.
In line 104, the authors present the results in a manner that does not read easily. Another way should be found to present the results here.
Line 174, the description of the sentence ‘the restaurant with 39% was the highest score’ is confusing.
Line 188 and 189, these sentences need rephrasing.
Page 7, end of table 2 – the question of when you choose a meal is in fact two questions that may have vastly different answers, which presents a problem in terms of the data collected here as well as the results presented.
Line 314 to 321 – there are aspects addressed here that have not been mentioned anywhere else in the manuscript, such as that the students received a salary from institutions, that they were distance learners, and others. This is a matter of concern, as these influences could affect the interpretation of the results.
The reviewer believes that the study results are of adequate quality and value, particularly in context of the growing nature of eating behaviours and consequent health as a result of food choices and lifestyles amongst humans.
Conclusion:
The reviewer believes that the conclusion is presented adequately and the worthiness of the research is evident, particularly in light of our never-ending fight against the global onslaught of diet-related illnesses amongst urban humans.
Author Response
Dear reviewer(s),
We appreciate the time devoted and the proposals you have made. The revised version of the manuscript has been modified according to your comments. Your detailed argumentation of each of the points covered in your review has allowed us to realize some deficiencies when transmitting the work done and improve it for the reader. We will be at your disposal to make any change that you deem necessary, resolve any questions or proceed with new revisions.
In the table below, all comments are answered. The lines in the “reviewer’s comment” and “authors’ reply” have been written with the “track changes mode in MS Word” activated.
Responses to Reviewer 2
Reviewer’s comment 1 |
Original version line(s): 36 |
remove the word ‘the’ To mitigate the outbreak, the epidemic has forced the international governments to im-pose harsh lockdown rules on their citizens as a precautionary measure against spreading the infection [3,4]. |
|
Author’s reply |
Revised version line(s): 38 |
To minimise the epidemic's risk, this has forced international governments to impose harsh lockdown rules on their citizens as a precautionary measure against spreading the infection [3,4]. |
|
Reviewer’s comment 2 |
Original version line(s): 38 |
replace the words ‘have been approved’ with ‘become a norm’ During the lockdown, study and work from home have been approved. |
|
Author’s reply |
Revised version line(s): 52-49 |
Another important aspect of the global lockdown is that all individuals' daily inter-actions, including studying and working, were converted to digital services. This orientation is associated with reducing any threats to human health, decreasing the pressure on health systems, and preventing the collapse of the healthcare system [5–8]. |
|
Reviewer’s comment 3 |
Original version line(s): 39 |
replace ‘bases have’ with interactions Therefore, almost all individuals' daily bases have converted …… |
|
Author’s reply |
Revised version line(s): 52-49 |
Another important aspect of the global lockdown is that all individuals' daily inter-actions, including studying and working, were converted to digital services. This orientation is associated with reducing any threats to human health, decreasing the pressure on health systems, and preventing the collapse of the healthcare system [5–8]. |
|
Reviewer’s comment 4 |
Original version line(s): 40 & 41 |
these sentences need rephrasing …… to digital services to save the life-threatening situation, relieve the tremendous pressure on the healthcare system, and prevent the system's collapse [5–8]. |
|
Author’s reply |
Revised version line(s): 52-49 |
Another important aspect of the global lockdown is that all individuals' daily inter-actions, including studying and working, were converted to digital services. This orientation is associated with reducing any threats to human health, decreasing the pressure on health systems, and preventing the collapse of the healthcare system [5–8]. |
|
Reviewer’s comment 5 |
Original version line(s): 40 |
consider changing ‘to save the’ to ‘ensure possible safety from’ ….. to save the life-threatening situation, …… |
|
Author’s reply |
Revised version line(s): 52-49 |
Another important aspect of the global lockdown is that all individuals' daily inter-actions, including studying and working, were converted to digital services. This orientation is associated with reducing any threats to human health, decreasing the pressure on health systems, and preventing the collapse of the healthcare system [5–8]. |
|
Reviewer’s comment 6 |
Original version line(s): 47 |
replace ‘rules’ with ‘interactions with FDAs’ However, these rules have negatively affected the general population's health status ….. |
|
Author’s reply |
Revised version line(s): 57 |
However, these interactions with FDAs have negatively affected the population's health and led to unhealthy behaviours, including sedentary behaviour and lack of physical activity [11,12]. |
|
Reviewer’s comment 7 |
Original version line(s): 48 |
check for consistency in the use of ‘behaviours’ and ‘behaviors’ ….. and led to unhealthy behaviours, including sedentary behavior and lack of physical activity [11,12]. |
|
Author’s reply |
Revised version line(s): 57 |
However, these interactions with FDAs have negatively affected the population's health and led to unhealthy behaviours, including sedentary behaviour and lack of physical activity [11,12]. |
|
Reviewer’s comment 8 |
Original version line(s): 52 |
add the words ‘of food’ after nutrient content For example, these applications can increase food consumption of high calories and low nutrient content, resulting in the risk of related diseases, including cardiovascular diseases, type 2 diabetes mellitus, and some cancers [14,15]. |
|
Author’s reply |
Revised version line(s): 61 |
For example, these applications can increase food consumption of high calories and low nutrient content of food, resulting in the risk of related diseases, including cardiovascular diseases, type 2 diabetes mellitus, and some cancers [14,15]. |
|
Reviewer’s comment 9 |
Original version line(s): 55 |
begin the sentence with ‘During lockdown, HungerStation… As a result of the forced experiment to remain at home, HungerStation, Marsoul, and Jahez are the most popular online FDAs in the Arabian Gulf region, especially Saudi Arabia. |
|
Author’s reply |
Revised version line(s): 65 |
During the lockdown, HungerStation, Marsoul, and Jahez are the most popular online FDAs in the Arabian Gulf region, especially in Saudi Arabia. |
|
Reviewer’s comment 10 |
Original version line(s): 57 & 58 |
this sentence needs total rephrasing; These applications have tremendous success helped during the pandemic and still are by facilitating consumer access to ready-to-eat meals and enabling food providers to keep operating [16]. |
|
Author’s reply |
Revised version line(s): 66 |
The success of these applications was obvious in delivering ready-to-eat meals to consumers and keeping food provider businesses viable [16]. |
|
Reviewer’s comment 11 |
Original version line(s): 70 |
replace ‘was enrolled’ with ‘took part’ A total of 3667 adult Saudi female volunteers was enrolled in this study …. |
|
Author’s reply |
Revised version line(s): 78 |
A total of 3667 adult Saudi female volunteers took part in this study, aged 18 to 55 years old and a citizen living in different regions. |
|
Reviewer’s comment 12 |
Original version line(s): 71 |
insert the word ‘and’ between ‘old’ and ‘a citizen’ …. with ages from 18 to 55 years old, a citizen living in different regions. |
|
Author’s reply |
Revised version line(s): 71 |
A total of 3667 adult Saudi female volunteers took part in this study, aged 18 to 55 years old and a citizen living in different regions. |
|
Reviewer’s comment 13 |
Original version line(s): 81 |
rephrase – ‘links that were distributed via emails and familiar… The online questionnaire was distributed links were placed on emails and familiar social media platforms such as WhatsApp, Twitter, Instagram, and Facebook. |
|
Author’s reply |
Revised version line(s): 98 |
The online questionnaire link was distributed via emails and familiar social media platforms such as WhatsApp, Twitter, Instagram, and Facebook. |
|
Reviewer’s comment 14 |
Original version line(s): 82 |
remove ‘design was’ The survey design was included a cover letter in Arabic and comprised the following two sections: ….. |
|
Author’s reply |
Revised version line(s): 99 |
The survey included a cover letter in Arabic and comprised the following two sections: sociodemographic characteristics and health information. |
|
Reviewer’s comment 15 |
Original version line(s): 84 |
end sentence after ‘questions’, start new sentence with ‘The first part…’ These sections consist of 25 closed-ended questions, the first part of the questionnaire was about demographics and health information … |
|
Author’s reply |
Revised version line(s): 101 |
These sections consist of 25 closed-ended questions. The first part of the questionnaire was about demographics and health information, including age group; marital status (single, married, divorced, widowed); education level (high school completion or lower, diploma, bachelors degree, postgraduate degree); occupation (employed, unemployed, student); monthly income (less than or equal 1000, more than 1000 to 5000, more than 5000 SR), and region of residence (West, Middle, East, North, South). |
|
Reviewer’s comment 16 |
Original version line(s): 90 |
‘height’ not ‘high’ Health status questions include high and weight to define the BMI level, suffering from any disease and obesity level. |
|
Author’s reply |
Revised version line(s): 107 |
Health status questions include height and weight to determine the BMI level [17], suffering from any disease and obesity level. |
|
Reviewer’s comment 17 |
Original version line(s): 90 |
‘determine’ not ‘define’ Health status questions include high and weight to define the BMI level, suffering from any disease and obesity level. |
|
Author’s reply |
Revised version line(s): 107 |
Health status questions include height and weight to determine the BMI level [17], suffering from any dis-ease and obesity level. |
|
Reviewer’s comment 18 |
Original version: 99 - on page 5 |
reference is made to the payment facility ‘Mada’, here it is referred to as ‘Made’ What are the payment methods you often use on delivery apps? - Responses were as follows: Upon delivery, Made, Apple pay, Visa. |
|
Author’s reply |
Revised version line(s): 115 |
Responses were as follows: upon delivery, Mada, Apple pay, Visa. |
|
Reviewer’s comment 19 |
Original version line(s): 101 |
Meal ingredients, Meal’s picture – check for consistency in the use of caps and punctuation marks What is the main factor that determines your food choice using delivery apps? with the following responses (The restaurant, Meal ingredients, meal's picture, Price, …. |
|
Author’s reply |
Revised version line(s): 116 |
What is the main factor that determines your food choice using delivery apps? with the following responses (the restaurant, meal ingredients, meal picture, price, delivery fees, … |
|
Reviewer’s comment 20 |
Original version line(s): 113 |
beginning bracket that should attach to the words ‘hard break’ In this study, the sample size was calculated using the Raosoft online calculator ( Raosoft Inc., Seattle, WA, USA, http://www.raosoft.com/samplesize.html, accessed on 3 January 2021). |
|
Author’s reply |
Revised version line(s): 130 |
This study calculated the sample size using the Raosoft online calculator (Raosoft Inc., Seattle, WA, USA, http://www.raosoft.com/samplesize.html, accessed on 3 January 2021). |
|
Reviewer’s comment 12 |
Original version line(s): 127 |
add the word ‘behavioural patterns’ to read ‘user’s behavioural patterns’ The chi-square statistical test is used to determine whether there is a relationship between demographic characteristics and the food delivery application user's pattern. |
|
Author’s reply |
Revised version line(s): 141 |
The chi-square statistical test is used to determine whether there is a relationship between demographic characteristics and the food delivery application user's behavioural pattern. |
|
Reviewer’s comment 22 |
Original version line(s): 135 |
replace ‘were’ with ‘had’ 58.3% of participants were undergraduate and postgraduate degrees; …. |
|
Author’s reply |
Revised version line(s): 150 |
From an educational point of view, 58.3% of participants were undergraduate and postgraduate degrees; while most participants were students (82.2%); the monthly income ranged from less than or equal to 1000 SR/month (80.5%) to more than 5000 SR (4.9%). |
|
Reviewer’s comment 23 |
Original version line(s): 136 |
the percentage of the ‘single’ criteria of sample is repeated in one sentence; …. while most participants were single and students (91.5% and 82.2%, respectively);…. |
|
Author’s reply |
Revised version line(s): 150 |
From an educational point of view, 58.3% of participants were undergraduate and postgraduate degrees; while most participants were students (82.2%); the monthly income ranged from less than or equal to 1000 SR/month (80.5%) to more than 5000 SR (4.9%). |
|
Reviewer’s comment 24 |
Original version line(s): 141 |
remove the word ‘were’ Eight participants were answered, … |
|
Author’s reply |
Revised version line(s): 157 |
Eight participants answered, not obese when asked in this survey, do you consider your-self obese? |
|
Reviewer’s comment 25 |
Original version line(s): 152 to 157 & the variables in Table 2 on page 6 |
the authors should take note to inform the reader of how HungerStation – Marsoul relates to HungerStation – Marsoul – Jahez, versus ‘HungerStation Marsoul’, particularly in Table 2, where it feels as if there is duplication – if the name of these APPS are indeed preceded by HungerStation in each example, then the authors should indicate it as such Summarised responses of using food delivery applications (FDAs) on the prevalence and dietary patterns among adult Saudi females have presented in Table 2. Generally, the most three popular FDAs were HungerStation - Marsoul – Jahez. Generally, the most three popular FDAs were HungerStation - Marsoul – Jahez in Saudi Arabia. While most apps were used during COVID-19 lockdown restrictions as follows, Hunger-Station-Marsoul were used by 30%, followed by HungerStation-Marsoul-Jahez (14.6%), HungerStation (14.2%), Marsoul (7.6%) and finally HungerStation - Marsoul - Jahez - To you (6.3%). |
|
Author’s reply |
Revised version line(s): 168 |
Summarised responses to using food delivery applications (FDAs) on the prevalence and dietary patterns among adult Saudi females have presented in Table 2. Generally, the three most popular FDAs were HungerStation, Marsoul and Jahez in Saudi Arabia. While most apps were used (%) during COVID-19 lockdown restrictions as follows: HungerStation and Marsoul (30%), followed by HungerStation, Marsoul and Jahez (14.6%), HungerStation only (14.2%), Marsoul only (7.6%) and finally by Hunger-Station, Marsoul, Jahez and To you (6.3%). |
|
Reviewer’s comment 26 |
Original version line(s): 161 |
‘using’ to replace ‘used’ …. participants used cash on delivery (21.7%), and the rest were used debit cards on delivery (Mada, Visa, Mastercard) (18.4%). |
|
Author’s reply |
Revised version line(s): 175 |
In comparison, less than one-quarter of participants used cash on delivery (21.7%), and the rest used debit cards on delivery (Mada, Visa, Mastercard) (18.4%). |
|
Reviewer’s comment 27 |
Original version line(s): 166 |
what is the purpose of the word ‘often’? The results showed that participants were classified based on their often food online orders …. |
|
Author’s reply |
Revised version line(s): 180 |
The results showed that participants were classified based on their food online orders within 1-2 days a month, a few days a month, a few days a week and daily with 37.6%, 36.3, 24.3% and 1.7%, respectively. |
|
Reviewer’s comment 28 |
Original version line(s): 201 |
what does ‘interesting feature’ mean? …. as the most interesting feature to use FDAs …. |
|
Author’s reply |
Revised version line(s): 215 |
Nearly half of the participants (49%) selected the delivery charge of FDAs compared to delivery speed (27.3%). |
|
Reviewer’s comment 29 |
Original version: Page 6 - Table 2 in the second, third and fourth column of the question |
‘On average, how many days do you order food through apps?’ there is an open line – remove; |
|
Author’s reply |
Revised version: Page 6 -Table 2 |
Table 2 was updated in terms of its structure. |
|
Reviewer’s comment 30 |
Original version: Page 7 - Table 2 |
what does the word ‘interesting’ imply in the most ‘interesting feature to use food delivery apps? When you choose a meal, do you read the description of the meal's ingredients, and can you determine the impact of the meal's content on the number of calories? |
|
Author’s reply |
Revised version: Page 7 - Table 2 |
What is the most important feature of using food delivery applications? |
|
Reviewer’s comment 31 |
Original version line(s): 240 |
begin sentence with ‘The use of Food Delivery Applications (FDAs) expanded greatly during the COVID-19…’ Food delivery applications (FDAs) shined during COVID-19 …. |
|
Author’s reply |
Revised version line(s): 256 |
The use of Food Delivery Applications (FDAs) expanded greatly during COVID-19 global lockdown restrictions. |
|
Reviewer’s comment 32 |
Original version line(s): 279 - 283 |
sentence too long One possible reason for this could be the marital status that may shed light on how easy to use FDAs by the single females who show a weak moral obligation in meal planning and home-meal preparation compared to married ones who most likely reflect more interdependent self-orientation with their families for the meal's planning and preparation [20]. |
|
Author’s reply |
Revised version line(s): 295 |
One possible perception could be marital status. The single women found it easy to use FDAs. Their responsibility toward meal planning and home-meal preparation appeared to be poor compared to married ones, who most likely reflect more interdependent self-orientation with their families for the meal planning and preparation [21]. |
|
Reviewer’s comment 33 |
Original version line(s): 286 |
Rather ‘Single women find it easy’ Singles are easy to use apps …. |
|
Author’s reply |
Revised version line(s): 302 |
Single women find it easy to use apps, while married women are mostly committed to preparing meals for their families. |
|
Reviewer’s comment 34 |
Original version line(s): 289 |
full-stop should be replaced by comma The total percentage of Adult Saudi Female participant who were obese (n= 733, 20%). single subjects and married were 17.2% and 2.8% respectively. |
|
Author’s reply |
Revised version line(s): 304 |
The total percentage of adult Saudi female participants who were obese (n= 733, 20%), single subjects, and married were 17.2% and 2.8%, respectively. |
|
Reviewer’s comment 35 |
Original version line(s): 291 |
rephrase ‘are well widespread’ is not correct One more finding of this study is that unhealthy eating patterns are well widespread among adult Saudi females by ordering from food delivery applications. |
|
Author’s reply |
Revised version line(s): 307 |
One more finding of this study is that unhealthy eating patterns are prevalent among adult Saudi females by ordering from food delivery applications. |
|
Reviewer’s comment 36 |
Original version line(s): 345 |
replace ‘did not’ with ‘were not’ In contrast, a total of 41.2% of the participants did not influence by social media. |
|
Author’s reply |
Revised version line(s): 359 |
In contrast, 41.2% of the participants were not influenced by social media. |
|
Reviewer’s comment 37 |
Original version line(s): 378 |
The manuscript is very well referenced using relevant, up-to-date references, however the reviewer is uncertain if the in-text manner of referring to authors (replacing it by a bracketed number as in line 378) is correct. Whereas [58] found it is not promising that providing calorie labelling on the menu can prevent obesity. |
|
Author’s reply |
Revised version line(s): 279 & 393 |
The references in text have been edited. |
|
Reviewer’s comment 38 on Literature Review: |
|
This reviewer felt that better explication of what knowledge is, what food knowledge is, what dietary knowledge is, and particularly what attitudes are, should be provided. The authors make reference to a concept referred to as ‘attitude lifestyle behaviours’. This reviewer could find no such concept. This is concerning, as ‘attitudes’ are a very specific concept with very specific dimensions and indicators to measure. |
|
Author’s reply |
Revised version line(s): 123 |
Knowledge has been removed from the manuscript because it was appeared one time in the questionnaire. |
|
Reviewer’s comment 39 |
|
In fact, when using Google Scholar, the reviewer found this exact paper available as follows: https://assets.researchsquare.com/files/rs-1467008/v1/2fb4dbb1-833c-4b38-9250-9b9177f8d0a4.pdf?c=1656577778 |
|
Author’s reply |
|
That version is not published anywhere but this one was for the preprint only. |
|
Reviewer’s comment 40 on Results and Discussion |
Original version line(s): 19 |
the authors indicate that the sample was between 18 – 24 years, however in table 1, a large proportion of the sample (23%) were UNDER 18 years old. |
|
Author’s reply |
Revised version: Page 4 - Table 1 |
It has been corrected in Table 1. The category age should be 18 years, >18-24, 25-34, 35-44, 45-55 |
|
Reviewer’s comment 41 |
Original version line(s): 104 |
the authors present the results in a manner that does not read easily. Another way should be found to present the results here. Does your mental state affect your food choice? (Yes) or (no) or (sometimes), when do you prefer to order food? (Morning, afternoon, evening). |
|
Author’s reply |
Revised version line(s): 140 |
Does your mental state affect your food choice? (yes, no or sometimes); when do you prefer to order food? (morning, afternoon, evening)
|
|
Reviewer’s comment 42 |
Original version line(s): 174 |
the description of the sentence ‘the restaurant with 39% was the highest score’ is confusing. Regarding the top four factors for determining the participants' meal choice through FDAs, the restaurant with 39% was the highest score, followed by delivery fees with 18.3%, the meal ingredients with 17.8%, and the meal price with 17.6%. |
|
Author’s reply |
Revised version line(s): 216 |
The top four factors affecting the participants' meal choice through FDAs were the restaurant with the highest score 39%, followed by delivery fees with 18.3%, the meal ingredients with 17.8%, and the meal price with 17.6%. |
|
Reviewer’s comment 43 |
Original version line(s): 188 & 189 |
these sentences need rephrasing. In contrast, one-quarter (20.5%) reported sometimes that it would encourage them to select lower caloric food choices, and only less than one-quarter of participants (18.1%) would pay more attention to meal's ingredients posted on the menu. |
|
Author’s reply |
Revised version line(s): 233 |
In contrast, 20.5% of participants reported that sometimes they read the description of the meal ingredients to select the lower caloric food choices, and approximately 18.1% of participants reported that they paid more attention to meal ingredients posted on the FDA menu. |
|
Reviewer’s comment 44 |
Original version: Page 7, end of Table 2 |
the question of when you choose a meal is in fact two questions that may have vastly different answers, which presents a problem in terms of the data collected here as well as the results presented. When you choose a meal, do you read the description of the meal's ingredients, and can you determine the impact of the meal's content on the number of calories? |
|
Author’s reply |
Revised version: Page 7 -Table 2 |
When choosing a meal, do you pay attention to the number of calories written on the restaurant menu? |
|
Reviewer’s comment 45 |
Original version line(s): 314 to 321 |
there are aspects addressed here that have not been mentioned anywhere else in the manuscript, such as that the students received a salary from institutions, that they were distance learners, and others. This is a matter of concern, as these influences could affect the interpretation of the results. |
|
Author’s reply |
Revised version line(s): 285 &331 |
It has been modified the word (salary) to ‘’monthly financial rewards from their institutions’’ |

Reviewer 3 Report
The Authors presented the paper on the influence of using food dietary applications on Saudi female dietary habits and preferences during the pandemic. The article is well elaborated and adequately structured, the results are mostly clearly demonstrated, and the reasoning is sound. However, I have a few comments that Authors may take into account to improve the paper:
1. The research questions and/or hypotheses are missing
2. The literature review highlighting the study's background and indicating the paper's scientific significance should be done more carefully.
3. The results presented in Table 3 (relationship between sociodemographic characteristics and selected questions…) could be described in the text in more detail.
4. The Authors could also provide the limitations of this research.
5. There is a footer under Table 1 that is not needed.
Author Response
Reviewer 3
Dear reviewer(s),
We appreciate the time devoted and the proposals you have made. The revised version of the manuscript has been modified according to your comments. Your detailed argumentation of each of the points covered in your review has allowed us to realize some deficiencies when transmitting the work done and improve it for the reader. We will be at your disposal to make any change that you deem necessary, resolve any questions or proceed with new revisions.
In the table below, all comments are answered. The lines in the “reviewer’s comment” and “authors’ reply” have been written with the “track changes mode in MS Word” activated.
Responses to Reviewer 3
Reviewer’s comment 1 |
Original version line(s): 60 |
The research questions and/or hypotheses are missing. |
|
Author’s reply |
Revised version line(s): 68 |
Has been done |
|
Reviewer’s comment 2 |
Original version line(s): 40 |
The literature review highlighting the study's background and indicating the paper's scientific significance should be done more carefully. |
|
Author’s reply |
Revised version line(s): 43 |
Has been done |
|
Reviewer’s comment 3 |
Original version line(s): |
The results presented in Table 3 (relationship between sociodemographic characteristics and selected questions…) could be described in the text in more detail. |
|
Author’s reply |
Revised version line(s): |
Has been done |
|
Reviewer’s comment 4 |
Original version line(s): 383 |
The Authors could also provide the limitations of this research. |
|
Author’s reply |
Revised version line(s): 459 |
The limitation has been added |
|
Reviewer’s comment 5 |
Original version line(s): 149 |
There is a footer under Table 1 that is not needed |
|
Author’s reply |
Revised version line(s): 165 |
the footer has been removed |

Round 2
Reviewer 1 Report
Thanks to the Authors for addressing reviewers' comments.